# Hand Hygiene Teaching Strategies among Nursing Staff: A Systematic Review

**DOI:** 10.3390/ijerph16173039

**Published:** 2019-08-22

**Authors:** María B. Martos-Cabrera, Emilio Mota-Romero, Raúl Martos-García, José L. Gómez-Urquiza, Nora Suleiman-Martos, Luis Albendín-García, Guillermo A. Cañadas-De la Fuente

**Affiliations:** 1Andalusian Health Service, Avenida del Sur N. 11, 18014 Granada, Spain; 2Red Cross School of Nursing, University of Sevilla, Avenida la Cruz Roja N. 1, 41009 Sevilla, Spain; 3Faculty of Health Sciences, University of Granada, Avenida de la Ilustración N. 60, 18016 Granada, Spain; 4Faculty of Health Sciences, University of Granada, Calle Cortadura Del Valle S.N., 51001 Ceuta, Spain

**Keywords:** hand hygiene, handwashing, infection control, nurses, nursing education

## Abstract

*Background*: Patient safety is a priority of any healthcare system, and one of the most effective measures is hand hygiene. For this, it is important that health staff have correct adherence and perform the technique properly. Otherwise, the incidence of nosocomial infections can increase, with consequent complications. The aim here was to analyze hand hygiene training and the effectiveness of different methods and educational strategies among nurses and whether they maintained correct adherence over time. *Methods*: A systematic review was conducted in the sources CINAHL (Cumulative Index to Nursing and Allied Health Literature), Dialnet, Lilacs (Latin American and Caribbean Health Sciences Literature), ProQuest (Proquest Health and Medical Complete), Medline, SciELO (Scientific Electronic Library Online), and Scopus. The search equation with Medical Subject Headings (MeSH) descriptors was “Nurs* AND (handwashing OR hand hygiene) AND clinical trial”. The review was performed following the recommendations of the guidelines of the Preferred Reporting Items for Systematic Reviews and Meta-Analyses (PRISMA) statement. *Results*: *n* = 17 clinical trials were included, with a total of 5747 nurses and nursing students. Strategies such as reminder sounds, practical simulations, videos, and audiovisual media improved handwashing compliance. Adherence overtime increased by up to 60%. The greatest effectiveness was related to the use of povidone–iodine, which reduced colony formation compared Hand hygiene teaching strategies among nursing staff: a systematic review to soap. *Conclusions*: The strategies that go beyond teaching techniques such as lectures may be more effective at increasing hand hygiene compliance. Combined approaches to learning/instruction improve user satisfaction by enabling self-management, flexibility, and repetition.

## 1. Introduction

In 2005, the World Health Organization (WHO) launched its “Clean Care Is Safer Care” campaign, an international initiative to promote patient safety [1]. This was followed in 2009 with “Save Lives: Clean Your Hands”, which is the main strategy program currently being promoted worldwide [2]. In accordance with these actions, health organizations around the world, such as the Health Foundation, are targeting health personnel to promote effective hand hygiene, thus enhancing patient safety and reducing the incidence of adverse effects such as nosocomial diseases [3]. The correct technique of handwashing is based in the following five steps: (1) Wet the hands and apply soap or use a hydroalcoholic solution. (2) Rub the hands together, following the order of palm to palm, with the back of the left hand to the right palm with interlaced fingers, repeating with the other hand, with palms together with fingers interlaced, with the backs of the fingers against the palms, with fingers interlocked. Clasp the left thumb with the right hand and rub in rotation. (3) Repeat with the left hand and the right thumb, rubbing the tips of the fingers in the other palm in a circular motion, going backwards and forwards and repeating with the other hand. (4) Rub for at least 20 s. (5) If the washing was with soap, rinse with water and dry [1].

Hospital-acquired infections (HAIs) pose major problems for health systems around the world. They are of multifactorial origin, but appropriate hand hygiene among health personnel is the most effective measure for preventing their propagation. Therefore, adherence to hygiene recommendations is of crucial importance [4] and has been reflected in many studies, which have demonstrated a clear relationship between the control of HAIs and proper hand hygiene [5].

Unfortunately, few studies on the training and adherence of health personnel to hygiene recommendations have been done, despite the obvious implications for patient safety [6]. In view of these considerations, it would be useful to investigate the degree of awareness among health staff in general, and nurses in particular, of these questions. It has been observed that although nurses may be perfectly acquainted with the underlying ideas of hand hygiene and may be willing to put them into practice (up to 94% agree with these statements), the technique is performed correctly by only 52% of nurses [7], and sometimes by even fewer. This gap between theory and practice raises serious concerns. Therefore, it is important to analyze training strategies and the degree of adherence of nurses to hand hygiene guidelines, taking into account that effective performance in this respect contributes greatly to preventing the spread of nosocomial infections [8].

Patient safety is a priority of any healthcare system, and one of the most effective measures is hand hygiene. For this, it is important that health staff have correct adherence and perform the technique properly. Otherwise, the incidence of nosocomial infections can increase, with consequent complications. For this purpose, this study was undertaken to examine the effectiveness of different methods and training strategies to increase handwashing compliance and to determine long-term adherence. The question for this review was, “Which educational strategies are most effective at improving hand hygiene technique (and/or compliance) among nurses and nursing students?”

## 2. Methods

### 2.1. Search Strategy

A bibliographic search was carried out in the sources CINAHL (Cumulative Index to Nursing and Allied Health Literature), Dialnet, Lilacs (Latin American and Caribbean Health Sciences Literature), ProQuest (Proquest Health and Medical Complete), Medline, SciELO (Scientific Electronic Library Online), and Scopus in accordance with the recommendations made in the Preferred Reporting Items for Systematic Reviews and Meta-Analyses (PRISMA) statement [9]. The search equations used were “Nurs* AND (handwashing OR hand hygiene) AND clinical trial” and its equivalent in Spanish. The descriptors of the search equation were taken from the Medical Subject Headings (MeSH) thesaurus.

### 2.2. Study Selection, Data Collection, Critical Review, and Level of Evidence

The inclusion criteria were clinical trials; analyzing handwashing techniques and the effectiveness of different methods used by nurses or nursing students; being published in English or Spanish during the period January 2008 to July 2018; and being related to the subject of the present study.

Studies focused on preventing nosocomial diseases among patients were excluded, as were studies not applied to human subjects and duplicated studies.

The search and study selection process was conducted by two members independently: a third member was consulted in cases of disagreement. First, the title and abstract were read, and then the paper selected was read in full. A reverse and forward literature search was made of the references cited in the selected studies.

A data coding manual was used. The following variables were obtained from each study: (1) author, (2) country, (3) year of publication, (4) study design, (5) interventions (educational strategies and effectiveness of different methods), (6) characteristics of the sample, and (7) rate of compliance (initial and over time).

The levels of evidence and degrees of recommendation used were those stipulated by the Oxford Center for Evidence-Based Medicine (OCEBM) [10]. The risk of bias was evaluated using a critical reading checklist for randomized clinical trials (CONSORT) [11].

## 3. Results

In total, 3939 articles were obtained from the literature search, of which 43 were duplicates and were hence excluded. After reading the titles and abstracts, a further 3922 were excluded because they did not meet the inclusion criteria, finally leaving 17 articles available for analysis. Figure 1 shows a flow chart with the selection process for the studies included.

The sample population was composed of 5747 nurses and nursing students. Five of the studies considered were based on a mixed sample of physicians and nurses, but the numbers belonging to each group were stated.

The nurses studied worked in the following areas: intensive care units (ICUs) (*n* = 6 studies), geriatrics (*n* = 2), and mixed nursing wards (*n* = 7). The other articles were carried out on nursing students. The details of each included article are provided in Table 1.

The risk of bias, assessed using CONSORT criteria [11], showed two points: an inadequate blinding of the outcome assessor (use of different people as outcome assessors) and no reports on whether the data analysts were blinded.

### 3.1. Characteristics of the Clinical Trials

The period of observation of the nurses from the start of the intervention to the measurement of the results obtained ranged from 1 to 24 months [13,14,15,17,18,19,21,28]. The average degree of adherence to hand hygiene guidelines prior to the intervention ranged from 6.8% to 66% [12,13,15,16,17,18,19,20,28]. After the interventions, this rose in all cases, by 18%–70% [13,14,16,17,18,19,20,28]. In certain studies, changes were maintained through the follow-up period (ranges from 16.1% to 51%) [14,16,17,20,28]: on the other hand, others authors found that the adherence over time decreased by 8.4% after 24 months [28]. The degree of compliance with hand hygiene in all selected studies was evaluated by external observers.

Significant variations in compliance were observed according to the nursing department in which the study was conducted. The highest levels of compliance were recorded in intensive care services. In ICUs, nurses showed a hand hygiene compliance only in high-risk contact situations (such as endotracheal suction). Intervention groups increased from 60% to 72.7% before patient contact and from 20% to 70% after patient contact. Regarding the duration of hand rubbing, rotational rubbing of thumbs increased 16.9% [28].

In addition, some authors showed a relationship between sociodemographic variables in relation to the compliance of handwashing in ICUs, with nurses with less experience showing greater compliance [18,28].

### 3.2. Educational Strategies of Healthcare Staff in Hand Hygiene

For the interventions, the “Five Moments for Hand Hygiene” recommendations of the WHO were taken as a reference. Most of the interventions formed part of professional training and consisted of promoting behavioral changes. The personnel concerned agreed to change their hand hygiene technique for one month, and the reported improvements in adherence to the new handwashing technique ranged from 9% to 20% [18,21].

There were also various educational interventions such as the use of sounds as a reminder, posters, practical simulations in workshops, practical scenarios, videos, and role-playing (simulations with different roles) [12,14,15,16,17,20]. According to Xiong et al. [22], an approach to training nursing students in hand hygiene consisted of providing mixed interventions based on elements such as lectures, videos, and role-play. These authors reported that learning and adherence improved with respect to self-directed training. Thus, after six months, 95% of the mixed-intervention group complied with hand hygiene recommendations, compared to 57.5% of the self-directed training group.

Another tool was training with audiovisual media through an e-learning methodology representing real clinical situations [27,28]. A study of nursing students reported that adherence to hygiene guidelines was greater among those who received audiovisual training rather than by the standard classroom approach after 8 weeks, achieving a greater adherence to good handwashing skills, performance, and knowledge [27].

### 3.3. Effectiveness of Different Methods

Only one of the studies reviewed assessed the hand hygiene techniques performed. Hand rubbing with hydroalcoholic solution was compared, in no particular order, to hand rubbing with a hydroalcoholic solution according to the standard seven-step technique and to handwashing with chlorhexidine solution following the same technique [23]. A 7.5% reduction in colony-forming units (CFUs) was obtained by handwashing performed according to the standard seven-step technique with the hydroalcoholic solution and also with the chlorhexidine solution. However, the group using the hydroalcoholic solution required the least time for the process, making this the best option.

In the last case, one study evaluated the use of soap versus a hydroalcoholic solution or povidone–iodine and recorded 88% of CFUs among the nurses who washed their hands with a hydroalcoholic solution [24].

### 3.4. Educational Materials and Feedback Concerning Hand Hygiene

Several studies observed that, apart from providing training, healthcare institutions also seek to maintain adherence via reminder mechanisms such as posters, placing gloves near the work area, and providing more hydroalcoholic solution dispensers [13,14,16,19]. The use of such reminder methods improved long-term (defined as “at least four months”) adherence to handwashing recommendations, from 24.1% to 60.6% of the staff addressed. Another reminder measure that was employed was to instruct patients in the importance of handwashing, in addition to the use of posters. This approach obtained results similar to those reported in other studies, but awareness of the issue was extended to patients as a parallel benefit [17,20].

Another study of this question analyzed an intervention based on hand hygiene reminder measures such as the prominent display of containers with hydroalcoholic solution located in strategic areas of the hospital and even fitted with acoustic signals such as tags with reminder beeps [12].

Others authors reported that better results were obtained when teaching was provided by means of audiovisual media, achieving a greater adherence to good handwashing skills, performance, and knowledge [27].

According to Huis et al. [14,15], in addition to appropriate training for nursing staff and the provision of sufficient resources for hand hygiene, interventions based on leadership and social support are important to promote long-term adherence to handwashing standards. In the case analyzed, compliance with hand hygiene standards rose from 20% to 50%, and this persisted for at least six months.

### 3.5. Prevention-Focused Training

Research has also been conducted into preventing the appearance of hand eczema, as this condition can dissuade nurses from putting hand hygiene into good clinical practice. Thus, one study described an intervention in which training was based on participatory working groups and complementary teaching materials. Although there were no statistically significant differences in the prevention of eczema, there was an increase in the use of resources to prevent it, such as moisturizers or cotton gloves [26].

In contrast, other authors [25] reported that prevention and even improvements were achieved by an intervention group on hand eczema among a population of geriatric nurses. The main intervention consisted of a training seminar on eczema and its prevention, together with individualized advice on skin protection. As a result, the prevalence of hand eczema fell from 26% to 17% in the intervention group.

## 4. Discussion

The systematic review described obtained data on 5432 nurses and 315 nursing students who took part in 17 studies on hand hygiene techniques. Despite the low number of clinical trials carried out in this area, with specific reference to nursing staff, the selected articles all described clinical trials presenting high methodological quality and low levels of bias, which corroborated their good internal validity [10,11].

Previous studies in ICUs have shown data on hand hygiene adherence similar to those reported in this review, from 29% [29] to 74% [30]. Although the results of this study showed that after the intervention, the duration of hand rubbing, and specifically the rotational rubbing of thumbs, increased, other studies found that all of the steps of handwashing were practiced minimally by the staff [31], which could be attributed to a high workload and a lack of time in an emergency. In addition, the association between hand hygiene adherence and related variables such as age, gender, experience, or a profession in the ICU supported the same results [32].

Various techniques have been described for instructing healthcare personnel with respect to hand hygiene. Meanwhile, others, such as role-playing, that have been used in educational contexts could also be considered, and the results obtained from the latter approach were in line with those reported here [33]. Training with audiovisual media considerably improved the outcomes achieved compared to traditional teaching methods, since it provided a visual representation of real situations of clinical care. Furthermore, the use of audiovisual media improved user satisfaction by enabling self-management, flexibility, and repetition. However, some authors advocated combined approaches to learning/instruction in the view that traditional and other techniques are complementary [34]. For example, in medical students, training methods through traditional instruction and the use of audiovisual equipment under similar conditions of time and content were found to achieve 12% greater adherence to good handwashing techniques [35].

Although methodical hand rubbing with an antiseptic solution using the seven-step technique has been shown to reduce the formation of CFUs [36], there remains controversy, as some researchers claim that no single technique is clearly superior to any other, since the results obtained may be affected by other factors, such as the thoroughness of performance or the time employed in the procedure [37].

The relationship between better hand hygiene and fewer cases of cross-infection has been well established [38]. Logically, intensive care services and operating rooms pay particularly careful attention to questions of hygiene, applying strict antisepsis programs such as the “Zero-ventilator-associated pneumonia (VAP)” program, which has been shown to achieve high levels of adherence [39].

Among interventions aimed at raising standards of hand hygiene, the approach most commonly taken in the training of health personnel is to emphasize the need to take necessary action where and when appropriate, as has been noted by the World Health Organization [2]. However, as observed above, levels of adherence are uneven, depending on the service, and so this measure alone is insufficient to ensure patient safety in this respect [40,41].

Education and adherence behavior should begin in academic institutions because they are optimal environments to encourage good habits [4], since knowledge among nurses and students seems to be deficient [42], maybe due to some barriers such as allergic reactions, lack of staff, or lack of awareness [43,44,45,46]. Although traditional methods provide the necessary notions for the development of practical skills, they do not ensure the acquisition of knowledge [47]. The approach for nursing students is to learn and become competent for professional life, and therefore sufficient time must be guaranteed to learn and develop appropriate attitudes and practices (since some studies identified time as a barrier related to adherence) [48,49]. In addition, new teaching methods should enhance independence, autonomy [50], and motivation [51] and guarantee the acquisition of theoretical principles and their innate implementation, increasing adherence to handwashing [51].

The best way to achieve this is multifocal teaching with mixed methodologies, because it is not completely clear which teaching methods are better than others [52]. However, some systematic reviews have shown that tailored interventions addressing the determinants of practice improve hand hygiene compliance [53,54].

Some authors believe that the synergy of stimuli, including training as a basic pillar, enhances adherence to hand hygiene recommendations. The strategy of applying a saturation of external stimuli can be very effective in the short to medium term, but the positive impact usually slows or even reverses in the long term, partly due to increasing tolerance or the normalization of overstimulation [55]. Moreover, in the workplace, this saturation should be accompanied by cultural awareness of the importance of patient safety, in the absence of which adherence to hygiene standards may decrease [56].

According to some authors, the use of posters and/or specific trainings to raise the levels of hand hygiene is not as effective as strategies based on seeking a change in habits. Apparently, better results are achieved by implementing gradual changes in guidelines for nurses, thus motivating them to maintain levels of compliance and fostering a working environment in which patient safety is paramount [8,56].

Finally, it is important to highlight the role played by managers and their leadership skills. The support that they provide is essential to achieve good handwashing in health professionals. Furthermore, it is very important that during the implementation process, managers retain focus to prolong adherence [57].

### Limitations

The present study presented certain limitations. First, due to the small number of studies that have been conducted on nursing staff and their compliance with hand hygiene standards, together with considerable heterogeneity within these research activities, it was not possible to perform a meta-analysis, and therefore our findings have limited external validity. Moreover, we did not investigate the reasons for nurses failing to maintain levels of hand hygiene despite their awareness of its importance. This question is probably related to motivational aspects and/or situations of overload and stress in the workplace [58]. Another limitation was related to the nature of the measurements (through external observation), which could affect the behavior of nurses, increasing their adherence to handwashing during the observation period (Hawthorne effect).

In view of these considerations, we suggest that future research in this area should address not only the degree of adherence to recommended techniques of hand hygiene, but also nurses’ motivation to do so and/or the workplace factors that may influence performance of these techniques.

## 5. Conclusions

It is essential to ensure the appropriate training of healthcare personnel in order to increase adherence to hand hygiene recommendations. In addition, strategies based on complementary stimuli should be adopted, since they improve adherence to handwashing by up to 70%. Finally, better results in increasing adherence to handwashing are obtained when traditional teaching methods are accompanied by the use of audiovisual media. The domination of skills is based on practice. Thus, it is essential to find strategies that go beyond the usual teaching techniques through the use of more innovative and flexible digital techniques.

## Figures and Tables

**Figure 1 ijerph-16-03039-f001:**
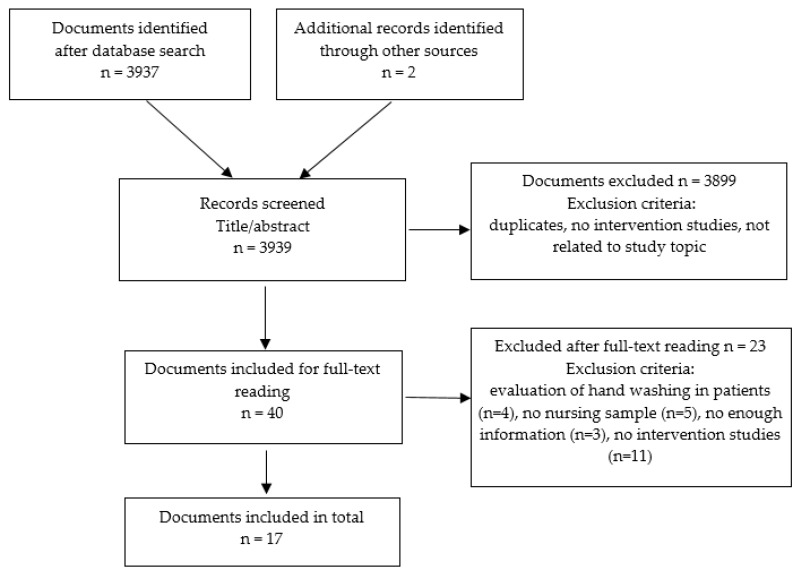
Flow chart of included studies.

**Table 1 ijerph-16-03039-t001:** Characteristics of included studies (*n* = 17).

Author, Country, (Year)	Design	Interventions	Sample	Hand Hygiene Compliance	Main Results	LE/GR
		Educational strategies				
Fisher et al., Singapore, (2013) [12]	Randomized controlled trial	HH compliance using ultrasound + audio reminders	*n* = 72 ICU nurses	-	Higher HH compliance after intervention of 6.8% (95% CI, 2.5–9.5)	1a/A
Ho et al., China, (2012) [13]	Randomized controlled trial by groups	CG: reception of intervention package (posters, talks, hydroalcoholic solution)IG 1: same + glove pack slightlypowderedIG 2: same + powderless gloves	*n* = 612 geriatric nurses CG *n* = 189 IG1 *n* = 180 IG2 *n* = 243	CG = 19.5% IG1 = 27% IG2 = 22%	Increase in HH compliance 1 month/4 months after interventionCG = 19.8%/21.6% IG1 = 59.2%/60.6% IG2 = 59.9%/48.6%	1a/A
Huis et al., Netherlands, (2013a) [14]	Randomized controlled trial	Adherence to two improvement strategies of HH Group led by leaders (GLD) (classic education) State-of-the-art strategy group (SASG): education, reminders, feedback, facilities and products, establishment of norms and objectives, social influence, and leadership	*n* = 67 nurses GLD: *n* = 20 SASG: *n* = 47	GLD: 19.1% SASG: 21.8%	Increase of HH adherence through social influence and enhanced leadership in HH improvement strategies Postintervention/follow-up (at 6 months): GLD: 34%/33% SASG: 18.6%/24.1%	1a/A
Huis et al., Netherlands, (2013b) [15]	Randomized controlled trial by groups	Strategy of HH compliance that was leader-directed CG: education, reminders, feedback, and orientation (led by leaders) IG: same as the last group + social influence and leadership (state-of-the-art wards)	*n* = 914 nurses, 67 wards CG: *n* = 402 IG: *n* = 512	CG: 20% IG: 22%	HH compliance rates improved from 22% (just before implementing strategies) to 47% (after the intervention) and to 48% (six months after). The vanguard group improved from 23% to 42% in the short term and 46% in the long term. Compliance in CG increased from 20% to 53% (short term) and remained at 53% in the long term	1a/A
Kukanich et al., United States, (2013) [16]	Randomized controlled trial	Improved HH in two outpatient healthcare clinics Outpatient clinic oncology (G1) Gastrointestinal specialist outpatient clinic (G2)Disinfectant gel and informational signs were introduced together as an intervention	*n* = 56 nurses G1: *n* = 41 G2: *n* = 15	G1: 11% G2: 21%	The frequency of HH improved significantly after intervention G1: 36% G2: 54%	1a/A
Martín-Madrazo et al., Spain, (2012) [17]	Cluster randomized controlled trial	5MHH to evaluate HH CG: - IG: training of HWs through teaching sessions, the implementation of hydroalcoholic preparations, and the installation of reminder posters	*n* = 198 nurses CG *n* = 99 IG *n* = 99		Overall baseline compliance level: 8.1% Increased adherence to HH: 21.6% At 6 months: CG: 3.6%IG: 16.1%	1a/A
Rodríguez et al., Argentina, (2015) [18]	Conglomerate randomized controlled trial	Improving HWs compliance with HH Shipping one time per month of an intervention: (i) leadership commitment, (ii) surveillance of materials necessary to comply with hand hygiene and alcohol consumption, (iii) use of reminders, (iv) a screenplay of the project, and (v) feedback	*n* = 468 ICU nurses	50%	A multimodal strategy was effective for HH compliance Handwashing after interventions was 70%	1a/A
Rupp et al., United States, (2008) [19]	Randomized clinical trial	Adherence in the use of alcohol-based hand gel Unit A: educational program, reminders of handwashing, and leaflets with questionnaires. After 12 months, introduction of hydroalcoholic gel Unit B: installation of hydroalcoholic solution containers inside and outside of each patient care room	*n* = 174 ICU nurses	Unit A: 47% Unit B: 38%	Increase in the use of alcohol-based hand gel at 31% in both units Unit A: ● After the educational program: 62% ● With hand gel available: 66% Unit B: ● Hand gel available: 74%	1a/A
Stewardson et al., Switzerland, (2016) [20]	Conglomerate randomized controlled trial	Control group (G1): observation of participants Improved performance feedback group (G2): observation + feedback (verbal comments, reminder of 5MHH) Improved performance feedback and participation group (G3): observation + feedback + reports and posters every 3 months	*n* = 67 nurses G1 = 21 G2 = 24 G3 = 22	G1: 66% G2: 65% G3: 66%	HH compliance increase from 65% to 77% G1: 73% G2: 75% G3: 77% Effect of the intervention: G1 = OR, 1.41 CI (1.21–1.63) G2 = OR, 1.61 CI (1.41–1.84) G3 = OR, 1.73 CI (1.51–1.98)	1a/A
von Lengerke et al., Germany, (2017) [21]	Randomized controlled trial	CG: training measures on “clean hands action” (adaptation of World Health Organization’s (WHO’s) “Cleaner Care Is Safer Care” program) IG: application of 29 patterns of behavior change	*n* = 572 ICU nurses CG: *n* = 367 IG: *n* = 205	CG: 55% IG: 54% (compliance with HH in 2013)	Increased adherence to HH through behavioral interventions in 2013 vs 2015 CG: +9% (95% CI, 5.1–11.8) IG: +16% (95% CI, 11.9–18.9)	1a/A
Xiong et al., China, (2017) [22]	Randomized controlled trial	CG: self-directed readings IG: education sessions, with lectures, videos, role play + 15–20 min of individual online supervision and feedback sessions after each class	*n* = 84 nursing students (*n* = 42 in each group)		The level of knowledge about HH increased by 15% in the intervention group	1a/A
		**Effectiveness of Different Methods**				
Chow et al., Singapore, (2012) [23]	Randomized controlled trial	Compared the effectiveness of 3 HH protocols Protocol 1 (P1): hand rubbing with alcohol covering all hand surfaces in no particular order Protocol 2 (P2): manual scraping with alcohol using the standard seven-step technique Protocol 3 (P3): washing hands with chlorhexidine using the standard seven-step technique	*n* = 60 nurses		In terms of daily care, alcohol hand rubbing covering all hand surfaces was the most effective intervention The effectiveness of the three interventions was shown to be equally effective Time spent on chlorhexidine HH was 79.7 s vs alcohol HH at 26 s	1a/A
Sharma et al., India, (2013) [24]	Randomized controlled trial	Comparison of 3 HH methods ● Group hand washing soap (G1) ● Alcohol solution group (G2) ● Povidone iodine group (G3)	*n* = 105 NICU nurses		Povidone–iodine scrub and alcohol hand rubbing were superior to plain soap hand washing In the groups using alcoholic solution and povidone, the measurement of colony-forming units was lower than in the group using soap. Mean reduction was 38.6%.	1a/A
		**Prevention-Focused Training**				
Dulon et al., Germany, (2009) [25]	Randomized controlled trial	Increase protective behavior through a skincare program reducing skin disease CG: training seminar IG: training seminar + advice on interventions and protection of the skin by instructors	*n* = 388 geriatric nurses CG: *n* = 242IG: *n* = 146	CG: 19% IG: 26%	No differences between groups in work behavior (prevalence post-intervention= 17% in both groups). In IG, increase in the use of moisturizers and hand disinfection instead of hand washing.	1a/A
Van der Meer et al., Netherlands, (2014) [26]	Randomized controlled trial	Effects of a multifaceted implementation strategy on behavior, behavioral determinants, knowledge, and awareness of HWs regarding the use of recommendations to prevent hand eczema CG: only brochure IG: education, participatory work groups, and role models	*n* = 1649 nurses CG: *n* = 773 IG: *n* = 876	CG: 10.3% IG: 7.3%	IG group was significantly more likely to report hand eczema CG: 9.7% IG: 11.3% The intervention had a positive effect on the frequency of HH, the use of a moisturizer, and wearing cotton gloves	1a/A
		**Training with Audiovisual Media**				
Bloomfield et al., UK, (2010) [27]	Randomized controlled trial	Effects of a computer-assisted learning module (IG) vs conventional face-to-face classroom teaching (CG)	*n* = 231 nursing students CG: *n* = 113 IG: *n* = 118	-	Computer-assisted learning was effective in teaching both the theory and the skill of knowledge of hand washing	1a/A
Jansson et al., Finland, (2016) [28]	Randomized controlled trial follow-up study	Four phases: (1) Simulation; (2) Orientation to mannequin capabilities; (3) Practical scenario; (4) Post-scenario debriefing session.	*n* = 30 ICU nurses IG: *n* = 15 CG: *n* = 15	IG = 40.8%	HH adherence in IG increased to 59.2% (6 months after the intervention) and decreased to 50.8% (24 months after)	1a/A

Note: CG = control group; GR = grade of recommendation; HH = hand hygiene; HWs = health workers; ICU = intensive care unit; IG = intervention group; LE = level of evidence; NICU = neonatal intensive care unit; 5MHH = five moments for hand hygiene.

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
