# Peer review of "Hand Hygiene Teaching Strategies among Nursing Staff: A Systematic Review"

_ijerph, 2019, doi:10.3390/ijerph16173039_

Round 1
Reviewer 1 Report
The authors have conducted a thorough search of the literature seeking consensus on effective hand hygiene education for nurses. The literature search was done in a clearly defined manner following accepted practice. The references were timely ranging in the last 10 years with the majority in the last 5 years. The article was overall easy to read and follow logically. Both the flow chart and the Table 1 were comprehensive and informative.
One issue in the table was the labeling of the prevalence for the Van der Meer et al., study with the notation (eczema). While this was latter explained in the text, this notation in the Table was not clearly defined thus creating confusion for the reader. Suggest either making a footnote to explain in the table or to remove the notation and keep the explanation in the text only.
Adding a section that described the actual interventions in more detail would have added value and interest to the manuscript. While this level of detail added the discussion section would have been able to refer more specifically to intervention techniques rather than in the generalities as it is currently written. For example, the authors state that audiovisual training had greater impact but the reader does not know what this training consisted of. Having this detail would help the reader in evaluating feasiblity to practice.
The authors mentioned the need to start hand hygiene education in nursing students, however neglected to give context on the type of education being used in nursing education. The current literature review only contained 315 nursing students, representing only 5% of the overall subjects in studies included in this study. The referenced citations to support the statements by the authors are not focused on nursing students but nurses in practice. Nursing students and nurses are not the same in education, experience and accountability to patients. Given that this research included research targeting nursing students, the authors need to include more discussion on the difference needs and characteristics of this population that warrant consideration in the valuable discussion of hand hygiene education.
Overall, well written without overreaching result conclusions. Informative and timely to current practice and addresses an ongoing challenge in nursing practice.
Author Response
Response to Reviewer 1 Comments
Dear Reviewer,
Thank you very much for reviewing the manuscript and your recommendations for improving it. Please find below and highlighted in yellow the response to each recommendation. All the changes in the manuscript have also been highlighted in yellow.
Point 1: One issue in the table was the labeling of the prevalence for the Van der Meer et al., study with the notation (eczema). While this was latter explained in the text, this notation in the Table was not clearly defined thus creating confusion for the reader. Suggest either making a footnote to explain in the table or to remove the notation and keep the explanation in the text only.
Response 1: The Word “eczema” has been removed from the Table and it is only in the text.
Point 2: Adding a section that described the actual interventions in more detail would have added value and interest to the manuscript. While this level of detail added the discussion section would have been able to refer more specifically to intervention techniques rather than in the generalities as it is currently written. For example, the authors state that audiovisual training had greater impact but the reader does not know what this training consisted of. Having this detail would help the reader in evaluating feasiblity to practice.
Response 2: We have included more information about the interventions in the results.
Point 3: The authors mentioned the need to start hand hygiene education in nursing students, however neglected to give context on the type of education being used in nursing education. The current literature review only contained 315 nursing students, representing only 5% of the overall subjects in studies included in this study. The referenced citations to support the statements by the authors are not focused on nursing students but nurses in practice. Nursing students and nurses are not the same in education, experience and accountability to patients. Given that this research included research targeting nursing students, the authors need to include more discussion on the difference needs and characteristics of this population that warrant consideration in the valuable discussion of hand hygiene education.
Response 3: We have included studies with nursing students. We have expanded the information about it in the discussion.

Reviewer 2 Report
Martos-Cabrera and colleagues have performed a systematic review of trials to assess education strategies for hand hygiene among nurses and nursing students. This is an important topic as they highlight in their background section – while we know that hand hygiene is one of the best ways to prevent the spread of infection (and is even more important in this age of multi-drug resistant organisms), we struggle with how to increase compliance among healthcare workers.
While a lot of work has been done to perform the review, I think more work needs to be done to organize the review to ensure that it attempts to answer a more focused clinical question. Otherwise, the manuscript resembles a review article rather than a systematic review. I suggest following the PRISMA statement guidance for creating systematic reviews: https://www.ncbi.nlm.nih.gov/pmc/articles/PMC2714657/
Generally, I think the methods need to be fleshed out more to clearly define which articles were included and which outcomes were looked at. For example, it is not clear why the study by Chow et al (Am J Infect Control. 2012 Nov;40(9):800-5) was included in the review, since that trial compared the effectiveness of different methods of hand hygiene, whereas the review was looking at different strategies for hand hygiene education among nurses.
The authors should begin with a clearly stated question, such as: Which educational strategies are most effective at improving hand hygiene technique (and/or compliance) among nurses and nursing students?
Then, define the interventions and outcomes that will be analyzed – of course, in a systematic review, there will be heterogeneity and likely multiple outcomes, but it would be helpful for the reader to understand specifically what was being looked at before presenting the results. It seems like the authors included articles that examined several outcomes:
Adherence to hand hygiene (most of the studies looked at this) Knowledge and attitudes of hand hygiene (e.g Bloomfield J et al, Int J Nurs Stud. 2010 Mar;47(3):287-94.) Hand hygiene technique (e.g., Sharma et al, 2013)It would be easier for the reader to split the articles into groups in the tables based on these outcomes (or focus on one outcome). And the authors may want to consider grouping interventions together with sub-headings, such as ‘Training with audiovisual media’.
One important thing to include in the table: describe how hand hygiene was measured in each study. It is well-known that measuring hand hygiene compliance is difficult. If hand hygiene compliance is measured using observers, this can affect the nurses’ behavior (i.e. people are more likely to wash their hands if they know they are being watched).
While the authors have stated that a meta-analysis was not done due to heterogeneity, there should still be a description of how outcome were measured, assessment of bias, etc.
More specific comments are below:
Abstract: The abstract does not describe the study well. It should clearly state that this is a systematic review looking at a certain outcome, then in the results, explain what the outcomes were.
Introduction: If the aim was to assess different teaching strategies to improve hand hygiene technique, then some background discussion should address what constitutes ‘good’ hand hygiene and whether this is associated with improved outcomes.
Figure 1: Describe in the methods what the ‘other sources’ were that yielded 2 additional articles.
Table 1:
It is not clear what the header ‘Prevalence’ means in the Table. Try to describe the results of the studies in a consistent fashion. Of course, the studies will have different outcomes, but perhaps start with increase in hand hygiene compliance in intervention vs. control groups.Line 78: Was the full text of 3,939 articles actually read? Or was the abstract read initially to decide whether to include in the study? I would think that many articles could be excluded simply by reading the abstract (and Figure 1 seems to imply that this was the case).
Line 106-107: It is not clear what ‘thumbs rotation time’ is.
Lines 108-110: For the previous paragraph in this section, the authors provide specific numbers. It would be good to provide numbers here to quantify ‘less experience’ and ‘older’, especially since these seem to be contradictory.
Lines 174-175: How did you assess bias in the studies? This needs to be described in the methods (instead of only referencing OCEBM guidelines)
Discussion/conclusions:
The authors should try to make more definitive conclusions. Which education strategies are not useful at improving adherence/knowledge/technique? If there are still knowledge gaps, what specific types of studies would they recommend?
I applaud the authors for the immense effort on this project. Thank for allowing me to review this manuscript.
Author Response
Point 1: While a lot of work has been done to perform the review, I think more work needs to be done to organize the review to ensure that it attempts to answer a more focused clinical question. Otherwise, the manuscript resembles a review article rather than a systematic review. I suggest following the PRISMA statement guidance for creating systematic reviews: https://www.ncbi.nlm.nih.gov/pmc/articles/PMC2714657/. Generally, I think the methods need to be fleshed out more to clearly define which articles were included and which outcomes were looked at. For example, it is not clear why the study by Chow et al (Am J Infect Control. 2012 Nov;40(9):800-5) was included in the review, since that trial compared the effectiveness of different methods of hand hygiene, whereas the review
Response 1: We have reviewed the PRISMA statement, the inclusion criteria of this review have been clarified.
Point 2: The authors should begin with a clearly stated question, such as: Which educational strategies are most effective at improving hand hygiene technique (and/or compliance) among nurses and nursing students?. Then, define the interventions and outcomes that will be analyzed – of course, in a systematic review, there will be heterogeneity and likely multiple outcomes, but it would be helpful for the reader to understand specifically what was being looked at before presenting the results. It seems like the authors included articles that examined several outcomes:
Adherence to hand hygiene (most of the studies looked at this) Knowledge and attitudes of hand hygiene (e.g Bloomfield J et al, Int J Nurs Stud. 2010 Mar;47(3):287-94.) Hand hygiene technique (e.g., Sharma et al, 2013) It would be easier for the reader to split the articles into groups in the tables based on these outcomes (or focus on one outcome). And the authors may want to consider grouping interventions together with sub-headings, such as ‘Training with audiovisual media’.
Response 2: We have included the question as recommended and we have clarified the results including a new subsection. Furthermore, the aim of each study is specified in the table.
Point 3: One important thing to include in the table: describe how hand hygiene was measured in each study. It is well-known that measuring hand hygiene compliance is difficult. If hand hygiene compliance is measured using observers, this can affect the nurses’ behavior (i.e. people are more likely to wash their hands if they know they are being watched)
Response 3: Adherence to hand hygiene in all selected studies was performed by observation. Although direct observation is considered the standard method for assessing hand hygiene compliance, it has certain limitations such as the possible Hawthorne effect. The observation modality of the selected studies has been included. It has also been included as a possible limitation.
Point 4: While the authors have stated that a meta-analysis was not done due to heterogeneity, there should still be a description of how outcome were measured, assessment of bias, etc.
Response 4: In the data collection section, the process for data collection has been included, as well as the instrument used for the evaluation of bias and methodological quality (CONSORT).
Point 5: More specific comments are below:
Abstract: The abstract does not describe the study well. It should clearly state that this is a systematic review looking at a certain outcome, then in the results, explain what the outcomes were.
Response: The abstract has been modified including that it is a systematic review and explaining the results with more clarity.
Introduction: If the aim was to assess different teaching strategies to improve hand hygiene technique, then some background discussion should address what constitutes ‘good’ hand hygiene and whether this is associated with improved outcomes
Response: What constitutes a good hand hygiene and its outcomes have been included in the introduction.
Figure 1: Describe in the methods what the ‘other sources’ were that yielded 2 additional articles.
Response: Other sources have been described in the methods sections.
Table 1: It is not clear what the header ‘Prevalence’ means in the Table. Try to describe the results of the studies in a consistent fashion. Of course, the studies will have different outcomes, but perhaps start with increase in hand hygiene compliance in intervention vs. control groups.
Response: The term has been changed for “hand hygiene compliance” because it is clearer.
Line 78: Was the full text of 3,939 articles actually read? Or was the abstract read initially to decide whether to include in the study? I would think that many articles could be excluded simply by reading the abstract (and Figure 1 seems to imply that this was the case).
Response: No, not all those articles full text were read. First, only titles and abstracts were read to assess the inclusion criteria. Full text reading was only done in 40 papers.
Line 106-107: It is not clear what ‘thumbs rotation time’ is.
Response: It refers to thumbs friction time during hand washing. The term has been clarified
Lines 108-110: For the previous paragraph in this section, the authors provide specific numbers. It would be good to provide numbers here to quantify ‘less experience’ and ‘older’, especially since these seem to be contradictory.
Response: The articles did not provided enough data. It has been clarified in the text.
Lines 174-175: How did you assess bias in the studies? This needs to be described in the methods (instead of only referencing OCEBM guidelines)
Response: More information has been included in the methods section.
Discussion/conclusions: The authors should try to make more definitive conclusions. Which education strategies are not useful at improving adherence/knowledge/technique? If there are still knowledge gaps, what specific types of studies would they recommend?
Response: Conclusions have been clarified. It has been included which kind of studies should be done.

Reviewer 3 Report
Very interesting article and suitable for the scientific community. I recommend acceptance
Author Response
Dear Reviewer,
Thank you very much for reviewing the manuscript.

Round 2
Reviewer 2 Report
Thank you for the opportunity to review the revision of the manuscript titled ‘Hand hygiene teaching strategies among nursing staff: a systematic review’. I appreciate your taking the time to address my comments and again applaud your efforts for this comprehensive systematic review.
Below are some comments/feedback on the revised manuscript:
Line 25: I don’t think ‘ratio’ is the correct word – unless you are actually looking at ratios.
Line 27-29: I’m not sure that the conclusions follow from the results. Summarize the conclusions in the discussion – such as pointing out that strategies that go beyond usual teaching techniques, such as lectures, may be more effective at increasing hand hygiene compliance.
Lines 39-41: the ‘5 moments’ that is discussed is more related to WHEN one should wash hands, rather than proper technique.
Lines 82-85: This is very helpful to the reader. However, I’m not sure that the term ‘outcomes’ is used correctly here (see below in comments about table)
Lines 87-88: This is also helpful, but then the authors do not mention in their results what risk of bias was found in the articles.
Line 91: delete ‘and full text’ since you did not read the full text of the 3,922 excluded articles
Table 1: I think the heading ‘Outcomes’ is misplaced. It seems like what is listed under that heading is interventions. The highlighted terms: Educational Strategies, Prevention-focused training, Training with audiovisual media are not outcomes – the outcome measured was hand hygiene compliance, knowledge of hand hygiene, etc.
What I recommend for Table 1 would be:
replace the word ‘Outcomes’ with ‘Interventions’. Re-order the articles so that all of the articles that are looking at Educational Strategies are grouped together, with a subheading, like this:Education Strategies
Article #1 description
Article #2 description
Etc.
Prevention-focused training
Article #1 description
Article #2 description
Etc.
Training with audio-visual media
Article #1 description
Article #2 description
Etc.
Lines 119-120: This is the only time that rotational rubbing of the thumbs is mentioned. You should explain the significance of this in the Discussion
Good luck with you submission!
Author Response
Response to Reviewer 2 Comments
Dear Reviewer,
Thank you very much for reviewing the manuscript again. Please find below and highlighted in yellow the response to each recommendation. All the changes in the manuscript have also been highlighted in yellow.
Point 1: Line 25: I don’t think ‘ratio’ is the correct word – unless you are actually looking at ratios.
Response: The word "ratio" has been removed.
Point 2: Line 27-29: I’m not sure that the conclusions follow from the results. Summarize the conclusions in the discussion – such as pointing out that strategies that go beyond usual teaching techniques, such as lectures, may be more effective at increasing hand hygiene compliance.
Response: Conclusions have been clarified and modified according to your recommendation.
Point 3: Lines 39-41: the ‘5 moments’ that is discussed is more related to WHEN one should wash hands, rather than proper technique.
Response: The information about hand washing technique and the steps to follow, has been included.
Point 4: Lines 82-85: This is very helpful to the reader. However, I’m not sure that the term ‘outcomes’ is used correctly here (see below in comments about table)
Response: The word “outcomes” has been changed to “interventions”.
Point 5: Lines 87-88: This is also helpful, but then the authors do not mention in their results what risk of bias was found in the articles.
Response: Risk of bias has been included in results.
Point 6: Line 91: delete ‘and full text’ since you did not read the full text of the 3,922 excluded articles
Response: It was deleted “and full text”.
Point 7: Table 1: I think the heading ‘Outcomes’ is misplaced. It seems like what is listed under that heading is interventions. The highlighted terms: Educational Strategies, Prevention-focused training, Training with audiovisual media are not outcomes – the outcome measured was hand hygiene compliance, knowledge of hand hygiene, etc. Replace the word ‘Outcomes’ with ‘Interventions’. Re-order the articles so that all of the articles that are looking at Educational Strategies are grouped together, with a subheading
Response: Table 1 has been modified based on recommendations and the word “Outcomes” has been replaced.
Point 8: Lines 119-120: This is the only time that rotational rubbing of the thumbs is mentioned. You should explain the significance of this in the Discussion
Response: This point has been explained in Discussion section.
